# Continuously Improving Mobile Manipulation with Autonomous Real-World RL

**Russell Mendonca** [1], **Emmanuel Panov** [2], **Bernadette Bucher** [2], **Jiuguang Wang** [2], **Deepak Pathak** [1]

[1]Carnegie Mellon University, [2]Boston Dynamics AI Institute

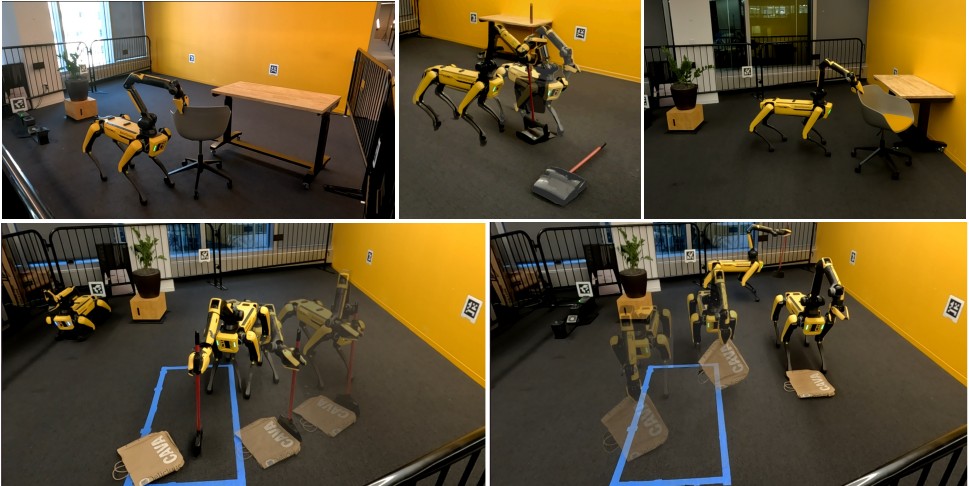

Figure 1: **Continual Autonomous Learning**: We enable a legged mobile manipulator to learn a variety of tasks such as moving chairs (top, left and right), righting a dustpan (top, middle), and sweeping (bottom) via practice in the real world with minimal human intervention.

**Abstract:** We present a fully autonomous real-world RL framework for mobile manipulation that can learn policies without extensive instrumentation or human supervision. This is enabled by 1) task-relevant autonomy, which guides exploration towards object interactions and prevents stagnation near goal states, 2) efficient policy learning by leveraging basic task knowledge in behavior priors, and 3) formulating generic rewards that combine human-interpretable semantic information with low-level, fine-grained observations. We demonstrate that our approach allows Spot robots to continually improve their performance on a set of four challenging mobile manipulation tasks, obtaining an average success rate of 80% across tasks, a **3-4**× improvement over existing approaches. Videos can be found at https://continual-mobile-manip.github.io/

**Keywords:** Continual Learning, Mobile Manipulation, Reinforcement Learning

## 1  Introduction

How do we build generalist systems capable of executing a wide array of tasks across diverse environments, with minimal human involvement? While visuomotor policies trained with reinforcement learning (RL) have demonstrated significant potential to bring robots into open-world environments, they often first require training in simulation [1, 2, 3, 4, 5, 6]. However, it is challenging to build simulations that capture the unbounded diversity of real-life tasks, especially involving complex manipulation. What if learning instead occurs through direct engagement with the real world, without extensive environment instrumentation or human supervision?

8th Conference on Robot Learning (CoRL 2024), Munich, Germany.

Prior work on real-world RL for learning new skills has been shown for locomotion [7, 8], and in manipulation for pick-place [9, 10, 11, 12] or dexterous in-hand tasks [13, 14, 15] in stationary setups. Consider a complex, high-dimensional system like a legged mobile manipulator learning in open spaces. The feasible space of exploration is much larger than in constrained tabletop setups. **Autonomous operation** of such a complex, high-dimensional robots often does not result in data that has useful learning signal. For example, we would like to avoid the robot simply waving its arm in the air without interacting with objects. Furthermore, even after making some progress on the task, the robot should not stagnate near goal states. While prior work has explored using goal cycles [16, 13, 17] to help maintain state diversity, this has not been shown for mobile systems. Such systems also need to learn more complex skills, involving constrained manipulation of larger objects and moving beyond pick and place, making **sample-efficient learning** critical. Finally, **reward supervision** using current RL approaches often requires physical instrumentation using specialized sensors [18, 19] or humans in the loop [20, 21, 22, 23], which is difficult to scale to different tasks.

Our approach tackles each of these issues of autonomy, efficient policy learning, and reward specification. We enable higher-quality data collection by guiding exploration toward object interactions using off-the-shelf visual models. This leads the robot to search for, navigate to, and grasp objects before learning how to manipulate them. We preserve state diversity to prevent robot stagnation by extending the approach of goal-cycles to mobile tasks and with multi-robot systems. For sample efficient policy learning, we combine RL with *behavior priors* that contain basic task knowledge. These priors can be planners with a simplified incomplete model, or procedurally generated motions. For rewards without instrumentation or human involvement, we combine semantic information from detection and segmentation models with low-level depth observations for object state estimation.

The main contribution of this work is a general approach for continuously learning mobile manipulation skills directly in the real world with autonomous RL. The main components of our approach involve: (1) task-relevant autonomy for collecting data with useful learning signals, (2) efficient control by integrating priors with learning policies, and (3) flexible reward specification combining high-level visual-text semantics with low-level depth observations. Our approach enables a Spot robot to continually improve in performance on a set of 4 challenging mobile manipulation tasks, including moving a chair to a goal with the table in the corner or center of the playpen, picking up and vertically balancing a long-handled dustpan, and sweeping a paper bag to a target region. Our experiments show that our approach gets an average success rate of about 80% across tasks, a **4× improvement** over using either RL or the behavior prior individually with our task-relevant autonomy component.

## 2   Related Work

**Autonomous Real-World RL:**   Previous work for real-world RL mostly involves either manipulation for table-top pick-place settings [9, 10, 8], in-hand dexterous manipulation [15, 13, 14] or locomotion behavior [24, 25, 8]. Approaches for automated resets needed for continual practice include instrumented environments [9, 10], forward-backward policies [26], graph structure of subtasks that serve as resets for one another [13, 14], or pre-trained, reliable reset policies [7]. For mobile manipulation, real-world RL has been limited to pick and place tasks [11, 12]. In our work, we extend the RL framework to learn challenging manipulation skills such as sweeping and moving chairs for a mobile system. Autonomous mobile systems should leverage the ability of the robot to move around to extend the effective reach of the robot and attempt manipulation tasks with large objects that are not possible on a table-top setup. For efficient learning on these complex tasks, we leverage behavior priors, which have some basic task knowledge. Moreover, task specification is a big challenge [27] for real-world learning. Current approaches often require physical instrumentation using specialized sensors [18, 19] or humans in the loop [20, 21, 22, 23], which is difficult to scale to different tasks. There has been some work on completely self-supervised learning systems with some extensions to robotics [28, 29], but these approaches are challenging to deploy on complex tasks due to intractability, underspecification, and misalignment. We extend the approach of using language goals and combining these with large-scale visual models [30], conditioned on open-vocabulary prediction [31, 32, 33], to obtain object states, which can be used to compute reward.

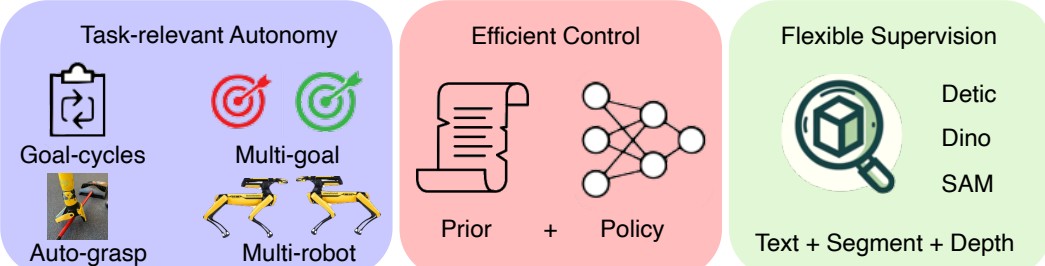

Figure 2: **Method Overview**: The main components of our approach for robots to continually practice tasks in the real world. **Left**: Task-relevant autonomy to ensure collection of useful data via object interaction, and maintaining state diversity via automated resets using multi-goal and multi-robot setups. **Center**: Efficient control by aiding policy learning with basic task knowledge present in behavior priors in the form of planners with a simplified model or automated behaviors. **Right**: Flexible reward supervision that combines human-interpretable semantic detection-segmentation information with low-level, fine-grained depth observation.

**Mobile Manipulation**   In the 2015 DARPA Robotics Challenge Finals, mobile manipulation solutions primarily relied on pre-built object models and task-specific engineering to enable mobile manipulation [34]. More recent work modularizing tasks into skill primitives and interacting with those primitives using flexible planners, including large language models, has enabled more generalization outside of pre-coded tasks [35, 36, 11, 37]. Imitation learning approaches to mobile manipulation enable joint reasoning over manipulation and navigation actions and generalize across broad sets of tasks [38, 39, 40, 41, 42]. However, imitation learning requires an expensive collection of expert trajectories. In contrast, RL methods can learn from experience without requiring extra human labor for each new task. Decomposing the action space over which the RL policy operates enables more tractable and efficient learning of long-horizon mobile manipulation skills [43, 44, 45, 46]. In our work, we move beyond tasks that involve picking and placing to instead learn skills that require coordination between the legs and arms, e.g., moving chairs or sweeping.

## 3   Continuously Improving Mobile Manipulation via Real-world RL

We design our approach to allow robots to autonomously practice and efficiently learn new skills without task demonstrations or simulation modeling, and with minimal human involvement. The overview of the approach we use is presented in Alg.1. Our approach has three components, as depicted in Fig 2: task-relevant autonomy, efficient control using behavior priors, and flexible reward specification. The first ensures the data collected is likely to have learning signal, the second utilizes signal from data to collect even better data to quickly improve the controller, and the third describes how to define learning signal for tasks. This allows learning difficult manipulation tasks, including tool use and constrained manipulation of large and heavy objects. Next, we describe each of these components in further detail.

---

**Algorithm 1** Autonomous RL for Mobile Manipulation

---

**Require:** Detection-segmentation models $M(.)$
**Require:** Behavior prior $P(.)$
 1: Initialize Data buffer $\mathcal{D}$, RL policy $\pi_\theta$
 2: Initialize task goal $\mathcal{G}_\mathcal{T}$ with goal object state $g_\mathcal{T}$
 3: Initialize trajectories per task $K$, horizon $H$
 4: **while** training **do**
 5:   **for** trajectory 1:K **do**
 6:     Approach object using Auto-grasp/nav
 7:     **for** timestep 1:H **do**
 8:       Use policy $\pi_\theta(.)$ and prior $P(.)$ for separate, sequential or residual control
 9:       Compute reward $r_t$ using $M(o_t)$
10:       Add $(o_t, a_t, o_{t+1}, r_t) \mapsto \mathcal{D}$
11:       Sample batch $\beta \sim \mathcal{D}$ to update $\pi$ via RL
12:     **end for**
13:     (optional) If distance$(x, g_\mathcal{T}) \leq \epsilon$, break
14:   **end for**
15:   Switch task goal $\mathcal{G}_\mathcal{T}$
16: **end while**

---

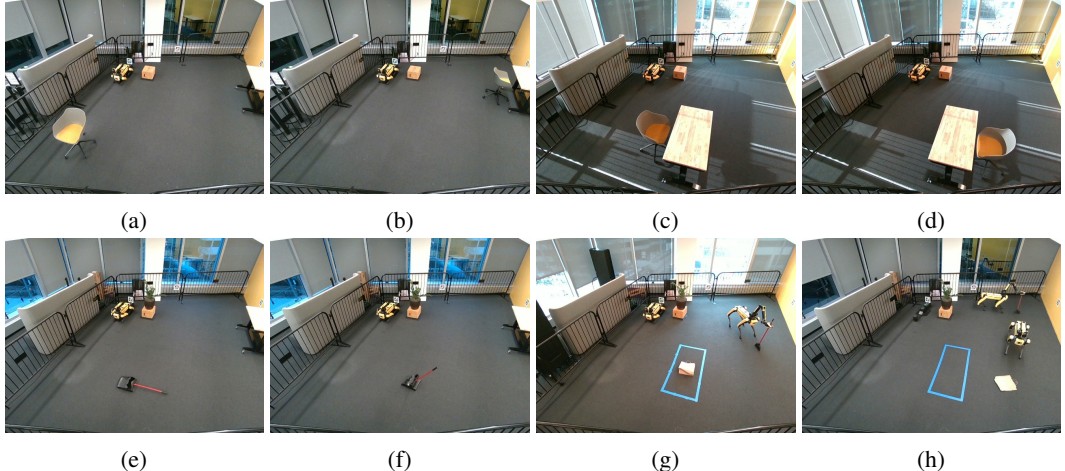

|       |       |       |       |
|-------|-------|-------|-------|
| (a)   | (b)   | (c)   | (d)   |
| (e)   | (f)   | (g)   | (h)   |

Figure 3: **Task Goals**: States that define goal-cycles for our 4 tasks - (a-b): Chair Moving with a corner table, (c-d): Chair Moving with a middle table, (e-f): Long Handled Dustpan Standup, (g-h): Sweeping

## 3.1 Task-Relevant Autonomy

**Auto-Grasp/Auto-Nav**: For safe autonomous operation, we first create a map by walking the robot around the environment. This map is used by the robot to avoid collisions during its autonomous learning process. To ensure data collected involves object interaction, every episode begins with the robot estimating, moving to, and/or grasping the object of interest for the task. The object state is estimated using detection and segmentation models along with depth observations, as described in section 3.3. The robot then navigates towards the object position using RRT* to plan in SE(2) space using the collision map, and optionally deploys the grasping skill from the Boston Dynamics Spot SDK depending on the task. This grasp is generated via a geometric algorithm that fits a grasp location with a geometric model of the gripper, scores different possible grasps, and picks the best one. We do not constrain the grasp type, or on which portion of the object the grasp is performed. This allows the robot to keep practicing regardless of which position or orientation the object might end up in as a result of continual interaction.

**Goal-Cycles**: To prevent robot stagnation near goal states, we set up 'goal-cycles' within tasks, which serve as automated task resets. We show the different goal states used in each of the 4 tasks we consider in Fig.3. In the case of the chair moving tasks (Fig.3: a-d), the robot alternates between goals that are far apart in the x-y plane, and for the dustpan stand-up task (Fig.3 e,f), the robot needs to pickup the fallen dustpan and vertically orient and balance it. For the sweeping task (Fig.3: g-h), we use a multi-robot setup for the goal cycle, where one robot holds the broom and needs to sweep the paper bag into the target region (denoted by the blue box), while the other needs to pick up the bag and drop it back into the region where it can be swept. Since we only need learning for the sweeping skill, the robot that picks up the bag runs the previously described auto-grasp procedure.

## 3.2 Prior-guided Policy Learning

**Incorporating Priors**: We enable efficient learning by leveraging behavior priors that utilize basic knowledge of the task. This removes the burden from the learning algorithm from having to rediscover this knowledge and instead focus on learning additional behavior needed to solve the task. For example, an RRT* planner with a simplified 2D model can help an agent move between two points in the x-y plane while avoiding obstacles. Starting with this prior, using RL can help the robot learn to recover from collisions and deal with dynamic constraints not represented in the model. Concretely, the prior is a function $P(.)$ that takes in an observation $o_t$ and produces an action $a_t$, similar to a policy $\pi(a_t|o_t)$. We can deploy the prior and the policy in the following ways:

1. *Separate*: Trajectories are collected independently using either the prior $\{P(a_0|o_0), \ldots, P(a_T|o_T)\}$ or the policy $\{\pi(a_0|o_0), \ldots, \pi(a_T|o_T)\}$. Instead of learning entirely from scratch, we incorporate the (potentially) suboptimal data from the prior into the robot's data buffer to bootstrap learning. Intuitively, the prior is likely to see a higher reward than a completely randomly initialized policy, especially for sparse reward tasks. We make no assumptions on the optimality of the prior, and bootstrap learning via incorporating its *data*. In practice, we first collect trajectories using the prior, to initialize the data buffer for training the online RL policy $\pi(.)$.

2. *Sequential*: In addition to providing data with better signal to the learning process, priors can reliably make reasonable progress on a task. This is because they often generalize well, for example, an SE(2) planner will make reasonable progress in moving a robot between any two points in the x-y plane, even when it performs constrained manipulation. We would need to sample many times from the prior to distill this information purely via the data buffer. Hence, a more direct approach is to utilize the prior along with the policy for control. We do this by sequentially executing the prior, followed by the policy. That is, trajectories collected in this manner take the form:

$$\{P(a_0|o_0), .., P(a_L|o_L), \pi(a_{L+1}|o_{L+1}), .., \pi(a_T|o_T).\} \tag{1}$$

Thus, the prior structures the policy's initial state distribution, making learning easier. The data collected by the prior is added to the data buffer, allowing the policy to learn from these transitions.

3. *Residual*: In certain cases, the prior might not be robust enough to deploy directly but nonetheless provide reasonable bounds on what actions should be executed. For example, for sweeping an object, the robot's base should roughly be in the vicinity of the trash being swept, but this does not prescribe what exact actions to take. Such a prior can be used residually, where a policy adjusts the actions of the prior at every time step before being executed. These trajectories take the form:

$$\{P(a_0|o_0) + \pi(a_0|o_0), \ldots, P(a_T|o_T) + \pi(a_T|o_T)\} \tag{2}$$

**RL Policy Training**: The RL objective is learn parameters $\theta$ of a policy $\pi_\theta$ to maximize the expected discounted sum of rewards $R(s_t, a_t)$:

$$J(\pi_\theta) = \mathbb{E}_{\substack{s_0 \sim p_0 \\ a_t \sim \pi_\theta(a_t|s_t) \\ s_{t+1} \sim \mathcal{P}(s_{t+1}|s_t, a_t)}} \left[ \sum_{t=0}^{T} \gamma^t R(s_t, a_t), \right] \tag{3}$$

where $p_0$ is the initial state distribution, $\mathcal{P}$ is the transition function and $\gamma$ is the discount factor. For sample efficient learning that effectively incorporates prior data, we use the state-of-the-art model-free RL algorithm RLPD [47]. RLPD is an off-policy method based on Soft-Actor Critic (SAC) [48], which samples from a mixture of data sources for online learning. Like REDQ [49], RLPD uses a large ensemble of critics and in-target minimization over a random subset of the ensemble to mitigate over-estimation common in TD-Learning. Since our observations consist of raw images, we incorporate the image augmentations added by DrQ [50] to the base RL algorithm.

### 3.3 Flexible Supervision via Text-Prompted Segmentation

For flexible reward supervision, we combine semantic high-level information from vision and language models with low-level depth observations. Each task is defined by a desired configuration of some object of interest, so we derive a reward function by comparing the estimated state of the object at a given time to this desired state (see Section 4 for task-specific details). To estimate the state of the object, we start by using an open-vocabulary detection model Detic [51] to obtain the bounding box corresponding to the object of interest. We then obtain the corresponding object mask by conditioning a segmentation model, Segment-Anything [30], on the bounding box. Finally, using depth observations and the calibrated camera system for either the egocentric or fixed third-person cameras, we get a point cloud. Although this estimation is noisy, we find it sufficient to enable learning effective control policies via real-world RL. This system is flexible enough to handle different objects of interest, such as the chair, long handled dustpan for vertical orientation, or the paper bag for sweeping. Full details on the prompts, detection and segmentation models, and reward functions for each task in the supplemental materials.

## 4 Experimental Setup

For our experiments, we run continual autonomous RL using the Spot robot and arm system in a playpen of about 6×5 meters, enclosed with metal railings for safety. The playpen is mapped before autonomous operation to ensure the robot stays within bounds and doesn't collide with the railings. The navigation aspect of task autonomy involves searching for objects of interest. Since the main focus of this work is on learning complex manipulation skills, we do not use learning for the search problem; instead, we rely on a fixed camera in the scene. In addition to this, we also use the 5 egocentric body cameras of the Spot while searching for objects.

The chair-moving task requires the robot to grasp a chair and move it between goal locations. We consider two variants, chair-tablecorner(Fig.3 a-b) and chair-tablemiddle(Fig.3 c-d). The latter is more challenging since collisions between the chair and table base are much more frequent

|  | Prior | Policy mode | Reward | Sparse |
|---|---|---|---|---|
| Chair-tablecorner | RRT* | Sequential | Chair-goal distance | False |
| Chair-tablemiddle | RRT* | Sequential | Chair-goal distance | False |
| Dustpan Standup | Scripted | Separate | Handle height | True |
| Sweeping | Distance constraint | Residual | Bag-goal distance | False |

Table 1: We list the choice of prior, how it is combined with the policy, how reward relates to the object state, and whether the reward is sparse.

and the robot has to operate in a much tighter space. The dustpan standup task involves lifting up the long handle of a dust-pan (Fig.3-e), and then vertically balancing it so that it can stay upright on its base (Fig.3-f). Sweeping involves two robots, where one of the robots holds a broom in its gripper and needs to use it to sweep a paper bag into a goal region (Fig.3-g). The other robot does not use learning, instead using the auto-grasp procedure to reset the paper bag by picking it up and dropping it close to the initial position(Fig.3-h). For each task, we specify success criteria for task completion, which corresponds to reaching the goal states in Fig.3. We list the choice of the prior, its combination with the policy, the state measurements used for reward, and reward sparsity in Table 1.

The observation space for RL policy training for all tasks consists of three 128X128 RGB image sources: the fixed, third-person camera and two egocentric cameras on the front of the robot. Additionally, we use the body position, hand position, and target goal. The action space for the chair and sweeping tasks is 5 dimensional, with base $(x, y, \theta)$ control and $(x, y)$ control for the hand relative to the base. The dustpan stand-up task is 3 dimensional, consisting of $(z, \text{yaw}, \text{gripper})$ commands for the hand, where the gripper open action terminates the episode. We use the same network architectures for image processing, critic functions, policy, etc., for all comparisons. Please see supplementary materials for more details on the full reward functions, success criteria, procedural functions for priors, hyper-parameters for learning, and network details.

## 5 Results

Our real-world experiments test whether autonomous real-world RL can enable robots to continuously improve mobile manipulation skills for performing various tasks. Specifically, we seek to answer the following questions: 1) Can a real robot learn to perform tasks that require both manipulation and mobility in an efficient manner? 2) Does performance continually improve as the robot collects more data? 3) How does the approach of structured exploration using priors along with RL, compare to solely using the prior, or using only RL? 4) How does the policy learned via autonomous training perform when evaluated in test settings?

**Task-relevant Autonomy:** Running the robot without auto-grasp or goal-cycles, with the full action space comprising base and arm movement to any position in the playpen does not lead to any meaningful change in task progress even over long periods of time. Further, such operation is unsafe since the robot arm can get stuck in the enclosure railings, or strike the wall in an outstretched configuration. Hence, all the experiments we conduct, including those for baselines, utilize the task-relevant autonomy component so that the robot can make some progress on the task.

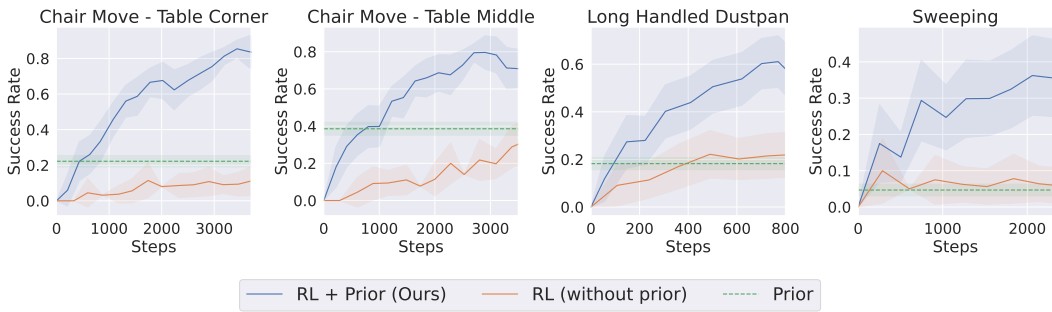

Figure 4: **Continual training improvement**: Success rate vs number of samples for ours, only RL and only prior. Note that we use our task-relevant autonomy approach with all methods. We see that our approach continuously improves with experience across tasks, learning much faster than RL without priors, and attaining significantly higher performance than just using the prior.

**Continual Improvement via Practice:** Given our task autonomy procedure, how effective is our proposed approach of combining real world RL with behavior priors, as opposed to using either only the prior or RL? From Fig.4, we see that our approach learns significantly faster than using only RL, and attains much superior performance than the prior, for each of the tasks. On the especially challenging sweeping task which involves tool use of the broom with a deformable paper bag, using only the prior or only RL leads to almost no progress, while our method is able to learn the task. Each robot training run takes around 8-15 hours, with the variation in time owing to different goal reset strategies across tasks and variance in how often the robot retries grasping objects for task-relevant autonomy. Hence, for fair comparisons across methods, we use the number of real-world samples collected to measure efficiency. The system also needs to be robust to many different factors in order to learn these tasks. The training area is exposed to sunlight, and the robot keeps collecting data and learning throughout the day with varying degrees of illumination. Object starting positions and grasps can vary widely, which affects the resulting object dynamics when practicing the task.

**RL without Prior:** For some tasks, using RL without the prior does improve in performance, but at a much slower rate than our method. Without the prior, RL often spends samples exploring parts of the state that are far from the goal. To illustrate this, we plot the average reward over each trajectory for the chair tasks (Fig.5). The reward for this task is of the form $-x + e^{-x}$, where $x$ is the distance of the chair to the goal position of the chair. The negative mean reward for RL without the prior implies that the distance $x$ to the goal is quite large, meaning that the robot is often far from the goal. On the other hand, since our method executes the prior and

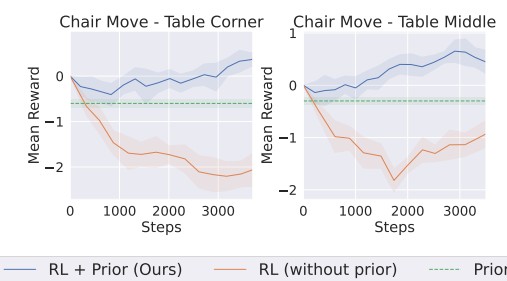

Figure 5: **Training mean reward**: Mean reward vs number of samples for the chair moving tasks. The negative average reward for RL without priors indicates that the robot is often far from the goal location.

policy sequentially for the chair task, our policy always starts out reasonably close to the goal, and can thus can pick up on high reward signal more often, leading to faster learning. We observe a similar pattern for the sweeping task, where using only RL leads the robot to wander around the playpen, greatly decreasing the likelihood of interacting with the paper bag and obtaining high reward.

**Prior without RL:** While the behavior priors are effective at bootstrapping learning, they are not sufficient on their own. This is because they do not adapt or learn from experience, and so keep repeating the same mistakes without improvement over time. We illustrate a qualitative failure example of the behavior prior for the chair moving task in Fig.6, where the robot following the RRT* planner runs into a collision state due to the simplified model being used. In contrast, our approach adapts the policy based on its experience to improve its performance, avoiding such collisions. For some tasks like sweeping the behavior prior is much simpler, only providing a constraint not to move too far away from the paper bag, which does not specify how the robot should sweep.

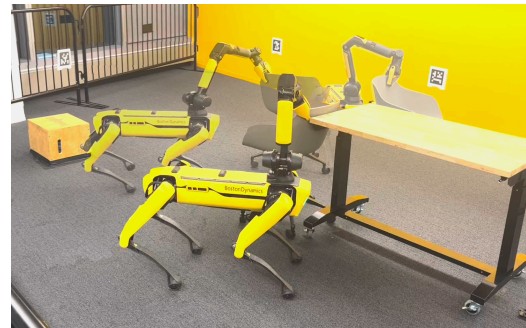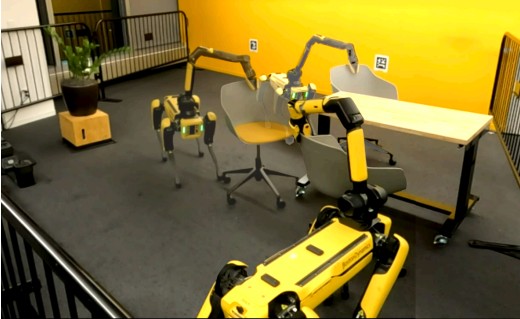

Figure 6: **Left**: The prior (RRT* with incomplete model) gets stuck in a collision with the table and is unable to recover as the planner does not have a model of chair-table interaction dynamics. **Right**: Our approach effectively recovers from collisions to complete the task.

**Final Policy Evaluation:** We evaluate the final policies obtained after autonomous, continual practice and find that our approach obtains an average success rate of 80% across tasks from Table 2. For comparisons between our method and using only RL, we evaluate models obtained with the same number of real world samples. For evaluation, we use the deterministic policy instead of sampling from the stochastic distribution, which is used during training.

Further, we set the initial state of the objects to be close to the opposite goal in the goal cycle. For instance, in the sweeping task, we initialize the paper bag roughly in the location shown in Fig.3-h. This is different from training, where the paper bag could end up in any location, and success is continually evaluated. We note that on the particularly hard task of sweeping, none of the other methods are successful, while our approach gets 80% success.

|  | **Ours** | Only RL | Only Prior | Offline RL |
|---|---|---|---|---|
| Chair-tablecorner | **100%** | 20% | 22% | 10% |
| Chair-tablemiddle | **80%** | 50% | 38% | 20% |
| Dustpan Standup | **60%** | 20% | 18% | **60%** |
| Sweeping | **80%** | 0% | 5% | 10% |

Table 2: **Evaluation Comparison**: The success rate of the final policy evaluated on different tasks. For evaluation, we use the deterministic policy instead of sampling from the stochastic distribution like in training. Our approach gets an average success rate of 80%, about 4× improvement over using only the prior or only RL.

**Prior Data Quality:** The behavior prior helps our approach in two ways, by structuring exploration for online learning, and also by providing higher quality data than random search, containing higher reward. To test the quality of the data obtained by the prior, we run offline RL on the dataset collected by the prior. This utilizes the reward of transitions to learn a policy, without any online rollouts. From Table 2, we see that on the chair and sweeping tasks, the behavior prior data quality is much worse, with an average success rate of 13%. The case of dustpan standup is notable since offline RL performs on par with our method, getting about 60% success. While the numerical performance is similar, there is a considerable qualitative difference in the behavior learned. Our approach learns strategies that are very different from the behavior prior, through exploration. This involves raising the robot's arm and dropping the dustpan, such that it lands upright. On the other hand, offline RL sticks close to the successful examples from the behavior prior generations.

## 6 Discussion and Limitations

We have presented an approach for continuously learning new mobile manipulation skills. This is enabled using task-relevant autonomy, efficient real-world control using behavior priors, and flexible reward definition. The current approach uses learning primarily for acquiring low-level manipulation skills after objects are grasped. Using automated procedures for navigation and search making use of a fixed third-person camera is a current limitation. This can be addressed by adding learning for the higher-level search problem too, which would allow the robot to rely just on its egocentric observations. This would allow learning in more unstructured, open-ended environments.

## 7    Acknowledgements

We thank Laura Smith, Murtaza Dalal, Ananye Agrawal, Kaiyu Zheng and Farzad Niroui for thoughtful discussions and their valuable feedback. This work was supported by the DARPA Machine Commonsense Program, Google Research Award and ONR N00014-22-1-2096.

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

# Appendix

## A  Videos

The main video summarizing our results can be found in result_video.mp4 in the zip folder. This depicts the robot performing each of the tasks we consider - moving the chair 1) with a table in the corner in the playpen, 2) with a table in the middle of the playpen, 3) picking up a dustpan and vertically orienting it such that it can stand up, 4) sweeping a paper bag into a target region. We also include timelapse videos which show how our approach adapts behavior over time.

## B  Policy Training

For our experiments we run DrQ implemented in the official RLPD codebase open-sourced by Ball et al. [47]. Since we run image-based real robot experiments, we use learning algorithm hyperparameters (including for the image encoders) from Stachowicz et al. [52], which deployed RLPD for race car driving. The observations are first encoded into a latent space (separately for the actor and critic), and the processed latent is used by the critic ensemble or the actor. Details of the architecture for each of these, in addition to hyperparameters for training is provided in Table 3.

We use both image and vector observations for learning. Each of these is processed by an image encoder or a 1-layer dense encoding for vector observations, and the corresponding latents are all concatenated together and then used as input for the actor or critic. Note that we use separate encoders for the critic and the critic. We use the architecture from Stachowicz et al. [52] for encoding each image source, without using any pre-trained embeddings, the network is retrained from scratch for each new experiment. There are 4 RGB image sources. The network encoders are provided with the last 3 frames for each image source, except for the goal image, since this remains fixed for the episode. The image sources are -

- Egocentric `front-left` image

- Egocentric `front-right` image

- Third-person fixed-cam current image

- Third-person fixed-cam goal image

We use (128,128) spatial resolution for the egocentric images, and (256,256) for the images from the third person camera. The latter uses a higher resolution since it is further away from the scene and objects appear smaller/less clear.

In addition, we have two vector observations -

- Body pose - We compute the (x,y,$\theta$) position of the robot body in the SE(2) plane relative to the calibrated playpen frame (calibration details in section D). The input to the network is 4 dimensional, consisting of $(x, y, \cos(\theta), \sin(\theta))$. We use $\sin, \cos$ transforms for the angle to avoid discontinuities in input, since $-\pi$ and $\pi$ represent the same orientation.

- Hand pose - 6-dof end effector orientation of the hand relative to the base position.

There are certain learning parameters that are tuned separately for each environment, which we list in Table 4. This was mainly to balance the exploration-exploitation trade-off for learning new behavior, and pertain to the weight placed on entropy maximization in DrQ (temperature and target entropy), or to handle sparse rewards (number of min Q functions). We use a maximum episode length of 16 for the chair and sweeping tasks, and 8 for the dustpan task, since it has sparse reward.

Table 3: Hyperparameters used in the experiments

| Category | Hyperparameter | Value |
|---|---|---|
| Training | Batch size | 256 |
| | Update to Sample Ratio | 4 |
| Actor/Critic | Actor learning rate | 3e-4 |
| | Critic learning rate | 3e-4 |
| | Actor network architecture | 2x256 |
| | Critic network architecture | 2x256 |
| | Critic ensemble size | 10 |
| Image Encoder | Layer count | 4 |
| | Convolution size | 3x3 |
| | Stride | 2 |
| | Hidden channels | 32 |
| | Output latent dim | 50 |

Table 4: Environment-tuned Hyperparameters

| Env | #MinQ | Temp LR | Init Temp | Target Entropy |
|---|---|---|---|---|
| Chair | 2 | 1e-4 | 0.5 | -2 |
| Dustpan | 1 | 1e-3 | 0.1 | -2 |
| Sweeping | 2 | 1e-4 | 0.1 | -4 |

## C  Rewards

### C.1  Detection-Segmentation

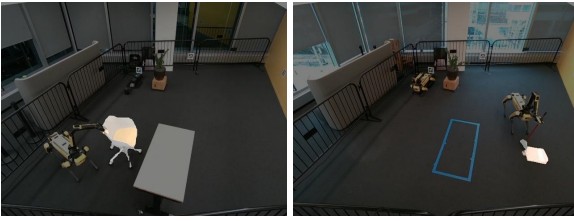

Figure 7: **Grounded SAM/Detic Visualization**: Visualization of the object masks obtained from Segment Anything for chair moving(left) and sweeping (right).

For each task, there is an object of interest, the state of which is used to compute the reward. We specify the object using a text prompt, which is used by the detection model to obtain a bounding box. This is then used to condition the Segment Anything [30] model to obtain a 2D object mask, as shown in Fig.7. For text-based detection we use either Grounding-Dino [53] or Detic [51]. For Grounding-Dino, we append the task-specific prompt to the list of class names in COCO [54] (to avoid cases of false positive detection), and we use Detic with `objects365` vocabulary class names. The task-specific text prompts we use are 'chair' for the chair tasks, 'red broom' for the dustpan standup task, and 'box.bag.poster.signboard.envelope.tag.clipboard.street_sign' for the sweeping task. The object of interest in the sweeping task is a paper bag being swept and we use many different possible matching text descriptions since it is detected as different classes due to its deformable nature. We list the detection model and the confidence threshold for a detection to be accepted for each task in Table 5.

Once we obtain object masks, we can obtain the corresponding object point-cloud using depth observations. Some detections are rejected based on estimated position, eg: if there is a detection of an object outside the playpen. This filtering is essential since the robot often picks up on known infeasible objects, eg: the box in the middle of the playpen, or some chairs outside the railings.

| Env | Detection Model | Confidence Threshold |
|---|---|---|
| Chair | Grounding-Dino | 0.4 |
| Dustpan | Grounding-Dino | 0.2 |
| Sweeping | Detic | 0.1 |

## C.2  Reward Function

**Chair-moving tasks**: For this task, we compute reward at every timestep of the episode. Given the estimated chair point cloud using the detection-segmentation system along with depth observations, we estimate the center of mass $x_t$ and the yaw rotation $w_t$. Given the goal position $g$ and orientation $g_w$ (extracted from the goal image), we compute position $x_{\text{diff}}$ and yaw difference $w_{\text{diff}}$ norms. Then the reward is given by :

$$r_{\text{position}} = -x_{\text{diff}} + e^{(-x_{\text{diff}})} + e^{(-10 \cdot x_{\text{diff}})}$$

$$r_{\text{ori}} = e^{(-w_{\text{diff}})} + e^{(-10 \cdot w_{\text{diff}})}$$

$$\text{Total Reward} = r_{\text{position}} + r_{\text{ori}}$$

**Dustpan Standup** In this task, it is difficult to provide reward when the robot is interacting with the dustpan, since the detection model fails to pick up on the dustpan from the third person or egocentric image observations. We can measure reward at the end of the episode (when the robot has released its grasp) to detect the dustpan and estimate the center of the handle $x_T$, and provide a large bonus if the height of the handle (z component of $x_T$) is above a set threshold. To prioritize faster task completion, we use an alive penalty of -0.1. The robot can terminate the episode earlier by releasing its gripper and letting go of the handle.

$$r_{\text{penalty}} = -0.1$$

$$r_{\text{bonus}} = 10 \text{ if } x_t \text{ height} \geq \text{thresh}$$

$$\text{Total Reward} = \begin{cases} r_{\text{penalty}}, & \text{if timestep } t < T \\ r_{\text{bonus}}, & \text{if end of episode, timestep } T \end{cases}$$

**Sweeping**: Similar to the chair task, we compute reward at every timestep of the episode. We estimate the point cloud of the paper bag, let its center of mass be denoted by $x_t$. The target region is a rectangle, denoted by $G_r$. Let $d(x, G_r)$ denote the distance from position $x$ to the closest corresponding point on the rectangle given by $G_r$. Then the reward is given by:

$$r_{\text{distance}} = -0.2 \cdot d(x_t, G_r) + e^{(-10 \cdot x_{\text{diff}})}$$

$$r_{\text{progress}} = 10 \cdot \max(0, d(x_{t-1}, G_r) - d(x_t, G_r))$$

$$r_{\text{bonus}} = \begin{cases} 10, & \text{if } d(x_t, G_r) = 0 \\ 0, & \text{else} \end{cases}$$

$$\text{Total Reward} = r_{\text{distance}} + r_{\text{progress}} + r_{\text{bonus}}$$

## C.3  Success Criteria

The results we show for continual improvement during training, as well as the evaluation of the final policies report success rate. Success is defined for an episode in the following manner:

- Chair tasks: Max reward in episode is above 1, implying the chair is very close to its target.

- Dustpan Standup: Episode ends with a reward of 10 (indicating the dustpan is standing up).
- Sweeping: Episode ends with a reward of 10 (paper bag is swept into the goal region).

## C.4 Priors

For the chair moving tasks we use RRT* for planning a path in SE(2) space with a simplified model that only has 2D occupancy of the top surface of the table, and is not aware of the chair, or robot-chair or chair-table interactions. This generates a set of way-points for the target position of the center of mass of the robot in SE(2) space, in global coordinates. We use coordinate transforms to convert these targets to be in the robot's body frame in order to use the same action space as the reactive RL policy. We are able to perform this computation since we know the robot's body position in global coordinates. Specifically, we have $W_{\text{body}} = W_{\text{global}} * T^{-1}$, where $W_f$ denotes the way-point with respect to frame $f$ and $T$ is the matrix transform of the robot body center of mass with respect

---

**Algorithm 2** Prior generation for Dustpan Standup

1: **Initialize** Prior data buffer $\mathcal{D}$
2: **Initialize** Uniform noise distribution $\mathcal{U}$ with limits :
   $(-0.1, -0.1, -1) \rightarrow (0.1, 0.1, 1)$
3: **for** $N = 1$ **to** Number of episodes **do**
4:    **Initialize** action list $\mathcal{A} = []$
5:    Set yaw hand rotation $\omega$ to either +0.5 or -0.5
6:    **for** $t = 1$ **to** episode len **do**
7:       Set vertical hand action $z$ to be either +0.2 or -0.2
8:       Add $(z, \omega, 0) + (n \sim \mathcal{U})$ to $\mathcal{A}$
9:    **end for**
10:   Add $(-0.2, \omega, 0) + (n \sim \mathcal{U})$ to $\mathcal{A}$
11:   Execute $\mathcal{A}$ on the robot, record observations, add to $\mathcal{D}$
12: **end for**
13: **return** Prior data buffer $\mathcal{D}$

---

to the global coordinates. For sweeping, the prior is simply to stay within 0.5m of the last detected location of the paper bag. For dustpan standup we use a simple procedural function to generate trajectories to create a prior dataset, which we detail in Algorithm 2

# D  Map Calibration

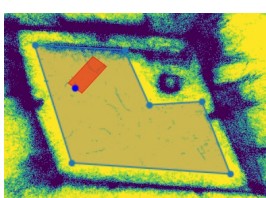

Figure 8: Collision map of the playpen used for safety and navigation. The table is added to this map when included in experiments.

We use the GraphNav functionality provided in the SpotSDK by Boston Dynamics for Spot robots for generating a map of the playpen. This involves walking the robot around with some fiducials (we use 5) in the arena. This needs to be performed only once, and is used to obtain a reference frame to localize the robot, which is useful to record body pose information and also to implement safety checks to make sure the robot is not executing actions that collide with the playpen railings. While Spot has inbuilt collision avoidance we implement an additional safety layer using the map to clip unsafe actions that would move the robot too close to the playpen railings. For navigation we use RRT* to plan in SE(2) space given the obstacles, using the collision map of the playpen as shown in Fig. 8. The red region denotes the estimate of the robot's position in the x-y plane, with the blue marking denoting its heading.

# E  System Overview

We use a workstation with a single A5000 GPU to run RLPD online, which requires about 20GB GPU memory, mostly owing to all the image inputs that need to be processed. The detection and segmentation models are run on cloud compute on a single A100 GPU. The fixed third person camera images from the realsense are streamed to a local laptop. Communication between the laptop, workstation and cloud server is facilitated via GRPC servers, and the main program script is run on the workstation, which also controls the robot. Commands are issued to the robot over wifi using the SpotSDK provided by Boston Dynamics.

