# OpenReview forum: "Continuously Improving Mobile Manipulation with Autonomous Real-World RL"
_robot-learning.org/CoRL/2024/Conference — CoRL 2024_

### Official Review · Reviewer_qcD5 · 2024-07-18
**Promising approach to a difficult problem in RL-based robotics**

**Originality:** 3
**Technical Quality:** 3
**Clarity Of Presentation:** 4
**Potential Impact:** 3
**Recommendation:** 3
**Confidence:** 4

**Review:**

**Quality, clarity, originality and significance**:

Two aspects of this paper appeal to me the most: (i) The leveraging of task-specific priors that help the robot navigate the search space effectively and helps it to focus on learning the actual manipulation skill (rather than learn how to reach target objects and learn the complete skill from scratch); and (ii) the mechanism for reward specification by utilizing off-the-shelf vision and language models. Training robots to learn manipulation skills with RL directly in the real world is challenging and this paper proposes a novel solution to this hard problem.

**Strengths**

1. The paper addresses a difficult problem in RL (training directly in the real world) and provides experimental evidence to show the effectiveness of the proposed approach.

2. Suitable experiments are performed in the real world involving manipulation skills that are not trivial to learn.

3. The paper is written well and is organized.

**Weaknesses**

1. The caption of Fig. 1, and the keywords prominently mention "Continual learning". For example, the caption of Fig. 1 states: "Continual Autonomous Learning: We enable a legged mobile manipulator to learn a variety of tasks ...". This gives the impression that the robot learns all 4 tasks with the same model with continual/lifelong learning, which I don't think is true.

2. All the manipulation tasks involve handling relatively large objects. It is not completely clear that the same level of performance can be maintained when handling smaller objects (e.g. spoon, an apple, etc.). At least one task with the manipulation of smaller objects would have further strengthened the paper.

3. Some experiments to show generalizability to different object variants (e.g. different chairs, brooms, etc.), lighting conditions, etc. would have been very nice.

**Quality Of The Limitations Section:**

3

**Questions For Rebuttal:**

1. Line 114: Please mention and cite the "geometric algorithm" that generates grasps.

2. Table 1: What is the reason for selecting different policy modes (sequential, separate, residual) for the different tasks? This should be mentioned in the paper.

3. Line 235: Each training run takes a long time (8-15 hours), making it very difficult to train several models or train in parallel (since the physical space cannot be simultaneously used by multiple robots). Can you comment on how your approach can be made more efficient in terms of training time?

4. Line 274: Please explain what is meant by "opposite goal".

5. Line 287: Which offline RL algorithm did you use?

6. How many independent trials were used to compute the reported results?

**Robotics Focus:**

4

**Summary Of Paper:**

The paper presents an approach to learning manipulation skills in the real world using a mechanism to collect useful training data, using task-specific prior knowledge to aid in training the controller, and flexible automated reward generation. Experiments presented in the paper show the effectiveness of the proposed approach in training a Spot robot to perform 4 real-world manipulation tasks.

**Summary Of Recommendation:**

The paper addresses a difficult problem in RL-based robotics and proposes an intuitive and effective solution. Multiple real-world experiments show that the proposed approach is effective and the work is presented in an organized and coherent manner. Although further experiments on generalizability could further strengthen the paper, I feel that the current results will benefit the robot learning community.

---

### Official Review · Reviewer_of2F · 2024-07-21
**Review for Submission 548**

**Originality:** 2
**Technical Quality:** 3
**Clarity Of Presentation:** 2
**Potential Impact:** 2
**Recommendation:** 3
**Confidence:** 3

**Review:**

Overall, while this paper does have a nice robotics demo, the main method is missing key detail and it is unclear the exact novelty in the proposed approach.


Strengths
+ This paper studies an important problem, continuous learning for new mobile manipulation skills.
+ The paper contains impressive robot demos.



Weaknesses
- To my knowledge, this work utilizes out-of-the-box RLPD and utilizes a prior to help guide the samples obtained online to better support the reinforcement learning. As these priors are hand-designed, can you comment on the key novelties of this technique compared to prior work and why this approach is better than prior work? For example, robots with intrinsic motivation strategies also have some prior to explore certain regions of the world. Could you also clarify the design of the prior for each task?
- The method section could be much improved with additional details about the priors, learning, and reward specifications.
- The mention of multi-robot early in the paper is misleading.
- The figures could be improved with better annotations. It is unclear what readers should focus on in Figure 3.
- The authors mention that this framework is for continuously learning new mobile manipulation skills. In the results section, it is unclear if the framework is attempting to learn a multi-task policy capable of many skills or if each task policy is separate. If the former, does the ordering of tasks being learned matter?

**Quality Of The Limitations Section:**

2

**Questions For Rebuttal:**

- Please address the weaknesses noted above.
- Could you comment on the feasibility of this approach? It seems 8-15 hours of training time for each task is relatively large.

**Robotics Focus:**

4

**Summary Of Paper:**

This paper presents a system for allowing for online learning in mobile manipulation without human supervision. The authors present several components in their approach, including task-relevant autonomy for collecting data, efficient control by integrating priors, and flexible reward specification. The authors find that combining their learning approach with a prior leads to the best performance when utilizing a Spot robot to perform various tasks.

**Summary Of Recommendation:**

The paper is missing details about the method and it is unclear what the novelties of this work are compared to those prior.

---

### Official Review · Reviewer_N2Ar · 2024-07-22
**Initial Review of Continuously Improving Mobile Manipulation with Autonomous Real-World RL Paper**

**Originality:** 3
**Technical Quality:** 4
**Clarity Of Presentation:** 4
**Potential Impact:** 2
**Recommendation:** 3
**Confidence:** 4

**Review:**

The paper addresses a relevant and open problem in RL – autonomous training and improvement in the real world. While there is a limited novelty in the components of the proposed framework, their combination and the application domain (robotic tasks, robotic system), to the best of my knowledge, is novel. Regarding the technical details, the methodology is sound and the descriptions provided in the main text and the supplementary material give enough information to understand the methodology and the experimental setup.

Strengths:

-	The framework enables real-world training that can learn meaningful policy in a reasonable time
-	The experiments show that the baselines (just priors or just RL) are inefficient or cannot learn a suitable policy, and the combination of components proposed by the authors significantly improves the results
-	The application area (the robotic tasks and the mobile manipulation with quadruped robot) is relatively unexplored by related work

Weaknesses:

-	Limited novelty – it combines existing components in a very general framework
-	Many decisions are task-specific, which makes it hard to apply this methodology easily on new tasks

**Quality Of The Limitations Section:**

3

**Questions For Rebuttal:**

-	There are three modes for combining the policy and the prior, is it necessary for a human to decide on the mode, or can it be automated in any way? For the tasks you used, did you decide based on the nature of the task, or did you check different modes and decide based on the empirical results?
-	You are using off-the-shelf models for detecting the environment state /calculating the reward and you use dense reward in most cases. How often do the off-the-shelf models fail on average in detecting the objects of interest and what happens in this case for calculating the reward in these timesteps?
-	How many runs (seeds) per approach did you do to calculate the mean reward/success rate, and what does the shaded area represent – please specify this for the figures.

**Robotics Focus:**

4

**Summary Of Paper:**

The authors propose a framework to train RL agents for mobile manipulation tasks directly on the real system. To achieve this, they propose 3 components: (1) task-relevant autonomy component that is tailored to enable task-specific data collection; (2) efficient control component that combines a behavior prior with a learning policy for online training and (3) flexible supervision component that integrates off-the-shelf detection/segmentation models for reward estimation. The authors use the framework to train and evaluate RL models on several tasks that involve mobile manipulation with quadruped robot in a simplified real-world environment. Their results show that the different components of their approach improve the sample-efficiency and the performance of the learned policy when compared with baselines that rely only on priors or only on RL training from scratch.

**Summary Of Recommendation:**

I recommend accepting the paper due to its application focus on real systems and the attempt for improving real-world RL, as well as because I think that future work in this field can be informed by the way the different components are integrated or can be used to improve the real-world LR training process.

---

### Official Review · Reviewer_zbZw · 2024-07-22
**Interesting, highly-engineered RL method for mobile manipulation**

**Originality:** 2
**Technical Quality:** 3
**Clarity Of Presentation:** 4
**Potential Impact:** 3
**Recommendation:** 3
**Confidence:** 5

**Review:**

- The authors propose an approach that, in theory, does not require human supervision for resets. This focus on autonomous learning in the real world without human oversight is an important and relevant topic for further research in the field.
- The paper is clearly written and easy to follow. The figures and tables are self-explanatory, and the authors do a good job explaining the robot experiments and setup.
However, there are several limitations to the approach:
- A significant amount of engineering appears to have been employed for the tasks where prior knowledge of task parameters, such as how to place the dustpan, is required.
- The priors and the experiment design do much of the heavy lifting. The approach seems heavily scripted for specific scenarios, which raises questions about its ability to generalize to other tasks or environments.
- Missing citations on RL for mobile manipulation:
[52] Honerkamp, D., Welschehold, T., & Valada, A. (2023). N2M2: Learning Navigation for Arbitrary Mobile Manipulation Motions in Unseen and Dynamic Environments. IEEE Transactions on Robotics, 39(5), 3601-3619.
[53] Jauhri, S., Peters, J., & Chalvatzaki, G. (2022). Robot learning of mobile manipulation with reachability behavior priors. IEEE Robotics and Automation Letters, 7(3), 8399-8406.
[54] Arm, P., Mittal, M., Kolvenbach, H., & Hutter, M. (2024). Pedipulate: Enabling manipulation skills using a quadruped robot’s leg. In 41st IEEE Conference on Robotics and Automation (ICRA 2024).

**Quality Of The Limitations Section:**

3

**Questions For Rebuttal:**

- The authors proposed three methods for incorporating priors into their approach: separate, sequential, and residual. A justification for their effectiveness in different contexts would be valuable. Why do particular priors work better in each environment, and how could this be predicted for future experiments on different tasks?
- Why does the performance of RL-only degrade in Fig 5?
- The authors can comment on how they could extend this approach to larger dimensional action spaces, where exploration would be more challenging and time-consuming.

**Robotics Focus:**

4

**Summary Of Paper:**

This paper addresses the problem of performing reinforcement learning (RL) in real-world mobile manipulation settings. The authors propose a learning approach that uses reward supervision via pre-trained models such as Detic and SAM. They also use priors relevant to the mobile manipulation tasks to speed up learning and show efficiency and performance benefits for four autonomous real-world robot learning tasks.

**Summary Of Recommendation:**

While the authors have shown useful real-world RL results for mobile manipulation, the overall approach isn't particularly novel. In its current form the paper also does not include an evaluation of why some behavior priors work better than others. However, extensive real-world experiments make me lean slightly toward acceptance.

---

### Author Rebuttal · Authors · 2024-08-09

Please see the replies to individual reviews for detailed responses to comments and questions. This file includes the updated paper and appendix incorporating suggested changes, referenced in the responses, as well as the original result video.

---

### Decision · Program_Chairs · 2024-09-04

**Decision:**

Accept

**Comment:**

The reviewers agree that the system and demo performance is great.  There is also general concern that the system may require a large amount of task-specific engineering.  One reviewer originally felt that the technical contributions were not described with enough clarity and detail, but was happy with the detailed response.  On balance, the feeling was that this paper provides a useful example of RL for manipulation that may inform and inspire future work.